# Cryostructuring of Polymeric Systems ^†^: Application of Deep Neural Networks for the Classification of Structural Features Peculiar to Macroporous Poly(vinyl alcohol) Cryogels Prepared without and with the Additives of Chaotropes or Kosmotropes

**DOI:** 10.3390/molecules25194480

**Published:** 2020-09-29

**Authors:** Ilya I. Kurochkin, Ilya N. Kurochkin, Olga Yu. Kolosova, Vladimir I. Lozinsky

**Affiliations:** 1A.A. Karkevich Institute for Information Transmission Problems of Russian Academy of Sciences, Moscow 127051, Russia; 2N.M. Emanuel Institute of Biochemical Physics, Russian Academy of Sciences, Moscow 119334, Russia; inkurochkin@gmail.com; 3Department of Chemical Enzymology, Chemical Faculty, M.V. Lomonosov Moscow State University, Moscow 119991, Russia; 4A.N. Nesmeyanov Institute of Organoelement Compounds, Russian Academy of Sciences, Moscow 119991, Russia; kolosova@ineos.ac.ru

**Keywords:** macroporous polyvinyl alcohol cryogels, chaotropic and kosmotropic additives, artificial neural network, texture convolutional neural network, Kylberg texture dataset, structural classification

## Abstract

Macroporous poly(vinyl alcohol) cryogels (PVACGs) are physical gels formed via cryogenic processing of polymer solutions. The properties of PVACGs depend on many factors: the characteristics and concentration of PVA, the absence or presence of foreign solutes, and the freezing-thawing conditions. These factors also affect the macroporous morphology of PVACGs, their total porosity, pore size and size distribution, etc. In this respect, there is the problem with developing a scientifically-grounded classification of the morphological features inherent in various PVACGs. In this study PVA cryogels have been prepared at different temperatures when the initial polymer solutions contained chaotropic or kosmotropic additives. After the completion of gelation, the rigidity and heat endurance of the resultant PVACGs were evaluated, and their macroporous structure was investigated using optical microscopy. The images obtained were treated mathematically, and deep neural networks were used for the classification of these images. Training and test sets were used for their classification. The results of this classification for the specific deep neural network architecture are presented, and the morphometric parameters of the macroporous structure are discussed. It was found that deep neural networks allow us to reliably classify the type of additive or its absence when using a combined dataset.

## 1. Introduction

Porous polymeric materials that are termed ‘*cryogels’* are known to be the gel systems formed via cryogenic processing (moderate freezing—frozen storage—thawing) of solutions or colloidal dispersions of appropriate precursors [1,2]. A characteristic feature of all the cryogel types is their macroporous morphology, since freezing of a feed system causes ‘liquid-to-solid’ phase separation with crystallization of the low-molecular weight solvent, the polycrystals of which act as porogens. Upon defrosting the system, the melting solvent fills the space of macropores [3,4,5,6,7]. The physicochemical characteristics and macroporous morphology of polymeric cryogels are stipulated by numerous factors. The main factors are the chemical nature of the precursors and their concentration in the initial solution, the cryoscopic properties of the solvent used, and the thermal conditions of the cryogenic processing [3,8,9,10]. Various polymeric cryogels are of substantial scientific interest as subjects for fundamental studies [1,3,4,5,10]. Also, these macroporous gels have high potential for many applications [1,3,4,9,11,12,13,14,15,16,17,18,19,20,21,22,23,24,25,26,27], such as their use in advanced materials for medicine, biotechnology, ecology, food technology, analytical systems, chemical catalysis, and so forth up to the implementation of special cryogels in construction engineering. 

Among the large number of the cryogels known nowadays, obviously some of the most popular examples are the physical (non-covalent) poly(vinyl alcohol) cryogels (PVACGs) that are formed as a result of freeze-thaw processing of concentrated solutions of this polymer [1,3,4,5,28,29,30,31,32,33]. The driving force of a such gelation phenomenon is the interchain H-bonding of the neighbouring OH-groups and, as a consequence, the formation of PVA microcrystallites that play the role of nodes within the supramolecular 3D network of the PVACGs [3,5,31,32,34,35,36,37]. The integrated characteristics of PVACGs are conditioned by the totality of the physicochemical properties of the gel phase proper in combination with the macroporous morphology of the material, i.e., by the amount, shape, size and interconnectivity of the pores [1,29,30,38,39,40,41,42,43,44]. In this respect, fool-proof quantitative methods for the morphometric analysis of the porosity characteristics of various PVACGs as well as for the classification of these indicators that are dependent on the gel-formation conditions are undeniably required. 

Thus, in or series of previous studies that dealt with the exploration of the structure/properties relations of different PVA cryogels, the evaluation of their macroporous morphology was accomplished via statistical methods and cluster analysis of the optical microscopy images recorded for thin sections of the respective PVACGs in an intact water-swollen state [43,44,45,46,47]. Such cryogels were prepared from aqueous or DMSO solutions of various PVA concentrations and molecular weights. In addition, other variables were the presence or absence of different soluble additives and the cryogenic processing conditions (e.g., chilling rate upon the freezing step, freezing temperature, heating rate during the defrosting step, the number of the freeze-thaw cycles, etc.). In all of these cases it turned out that it was possible to trace only certain selected interrelations between the physico-mechanical or thermophysical characteristics of the respective PVACGs and the morphometric parameters of their porous structure. To assess their macroporous morphology, images of cryogels were classified. The classes were determined based on the physico-mechanical or thermophysical characteristics of such PVA cryogels. In turn, the classification itself was based on the image parameters obtained by image processing and statistical methods. The size of the pores and their amount were selected as significant parameters. Dimensionality reduction methods, such as the principal component method, were used to identify the significant parameters. However, a small number of the images did not allow for the full use of classification methods, such as decision trees, neural networks, etc. 

Nowadays, deep convolutional neural networks or deep neural networks are widely used in the field of image classification [48,49]. These approaches ensure successive results that are often superior to other classification methods. According to the operation principle, deep neural networks can be classified as a class of methods based on signal processing. 

Deep learning refers to the area of neural network training with a large number of layers where classical methods of neural network training become unusable. The most significant problem in learning deep neural networks, or the so-called Fundamental Deep Learning Problem, is the rapid growth or damping of the error in the back propagation process when using standard sigmoid activation functions such as sigm(x), tanh(x), and others [50]. Besides, deep neural networks have a very large number of parameters (10^8^–10^10^) [51], which determine the relevance of developing of better methods for configuring them.

The most popular approach to training a deep neural network is to use the stochastic gradient descent method (SGD) [51]. Finding the error gradient on the entire set of input vectors can often require a significant amount of calculations, so in practice, gradients are calculated in batch mode, sequentially typing several elements of the training set into a batch where the average value of the gradient vector is calculated and the gradient descent step is performed. The gradient descent algorithm can be significantly improved by using adaptive approaches that involve dynamically adjusting the learning speed. These algorithms include Adagrad [52] and Adam [53]. The Adam method calculates the values of the first and second moments of the error gradient vector during training and adjusts the gradient descent step based on these values. It is also possible to improve the convergence of deep neural network learning algorithms by using alternative activation functions such as ReLU, ELU, and others. This made it possible to train deep neural networks more successfully than using sigmoid functions, due to the absence of the effect of gradient damping when the argument value increases [54]. In particular, unlimited top-down activation functions allow one to train deep neural networks without using layer-by-layer pre-training techniques. Besides, they are more computationally efficient, since there is no need to calculate exponential functions.

Deep neural networks require a large amount of data for training. Preparing data for neural network training is quite a time-consuming process, and in some cases it is impossible to obtain a large enough amount of data because of economic or other reasons. One of the most common solutions to the problem of a small sample of training data is to pre-train a neural network on data with identical significant features [55,56]. In the field of image classification, this approach can also be applied, since the lower convolutional layers of the neural network extract the simplest features, such as inclined straight lines, curved lines, and colour gradients. These features are fairly universal and can be successfully applied to classify images in other sets.

There are several approaches to pre-training a deep convolutional neural network. One of them is to train the network in several stages. At the first stage, the network is trained on the task of classifying a large set of images. The most popular texture image datasets are KTH-TIPS, Brodatz, DTD, and Kylberg Texture Dataset [57]. At the second stage, the network’s classifier is replaced with a new one, after which the network is fine-tuned on the target set of images. A much smaller gradient descent step is used, and sometimes the approach is used to fix the already configured weights of the lower layers of the network for several initial training epochs, after which the entire network is trained as a whole. In addition, pre-training makes sense when the target set of images is small. This approach allows one to increase the generalization capacity of the network and, as a result, the quality of classification even in the case of a large target set of images [56].

The similarity of images in the large reference set (dataset) and the target dataset also affects the effectiveness of the network pre-training process. However, the use of reference data sets allows one to weaken the influence of the specificity of the target dataset. This observation creates the prerequisites for the use of large sets of general-plan images in specific tasks, including the classification of texture images. 

The subjects of our study were PVA cryogels prepared from aqueous solutions of the polymer that either did not contain any foreign solutes or included the additives of organic chaotropes or kosmotropes [58]. The former ones were non-ionic urea (URE) and ionic guanidine hydrochloride (GHC); the latter ones were the non-ionic trehalose (THL) and the zwitterionic hydroxyproline (HYP). It was shown in our earlier study [59] that the influence of the above-listed compounds on the PVA cryotropic gel-formation and on the properties of the resultant cryogels has a multifactor character. Therefore, the structure/property correlations revealed with the aid of parameters of the images obtained using statistical methods were rather ambiguous. Besides, all feed solutions in the previous study were frozen at the identical temperature of −20 °C; in the present research, the cryogenic processing temperature (*T*_cp_) range was widened, since *T*_cp_ is the major factor capable of affecting the amount and size of the porogens, which is ice in our case. Fabrication of cryogels at various freezing temperatures resulted in a considerable increase in the data volume involved in the mathematical treatment of the images recorded with optical microscopy for the thin sections of PVACGs of our interest and allowed drawing more general conclusions on the structure/property interrelations for such porous polymeric materials. The main goal of this work is to confirm the hypothesis that a set of PVACG images can be classified using deep neural networks.

## 2. Results and Discussion

### 2.1. Physico-Mechanical Properties of PVACGs Prepared at Different Freezing Temperatures and in the Absence or the Presence of Chaotropic or Kosmotropic Additives

As revealed earlier, PVA cryogels could be formed and their physico-mechanical characteristics reliably measured with the instruments at our disposal (Section 3.4) when the amount of such chaotropic substances, such as the non-ionic URE and the ionic GHL introduced in feed polymer solution, did not exceed 0.5 and 0.3 mol/L, respectively [59]. Otherwise, the resultant PVACGs turned out to be very weak. It is evident that this effect is due to the PVA-PVA H-bonding inhibition induced by similar chaotropes. On the other hand, kosmotropic additives capable of promoting the formation of H-bonds, the non-ionic THL and the ionic HYP in our case, exhibited pronounced strengthening (in relation to PVACGs) effects commencing from about 0.5 mol/L content in the PVA solutions to be gelled cryogenically [59]. Therefore, the concentrations of the chaotropic or kosmotropic additives in the present study were as follows: 0.05–0.50 mol/L for URE, 0.05–0.30 mol/L for GHC, and 0.1–1.0 mol/L for THL and HYP. All of the PVA cryogel samples, i.e., the additive-free and the chaotrope- or kosmotrope-containing ones, have been prepared at four minus temperatures, namely at −20, −25, −30, and −35 °C, in order to reveal the impact of the cryogenic processing conditions on the rigidity, heat endurance, and macroporous morphology of these gel matrices. Herewith, the temperature range was stipulated by the following factors. On one hand, at *T*_cp_ as high as −20 °C, the PVA solutions that contained a large amount of THL or HYP, e.g., 1.0 mol/L, did not freeze at all because of the undercooling effects. On the other hand, the gel strength of all PVACGs formed at *T*_cp_ as low as −35 °C decreased sharply, so their fabrication and study from our viewpoint was unreasonable. 

The plots in Figure 1 show the dependences of the compression Young’s modulus (*E*) for the prepared PVA cryogels on the additive’s concentration (Figure 1a,c,e,g) and on the cryogenic processing temperature (Figure 1b,d,f,h). It is thought that such a mode for the results presented allows one to clearly trace the main effects associated with the chemical nature of the low-molecular additives and their amount in parallel with the influence of freezing conditions on the rigidity of the resultant PVACGs. Furthermore, for the convenience of data juxtaposition, all of the plots in Figure 1 are given with identical scales for the respective axes. 

First of all, the very similar character of the curves in Figure 1a,c related to the cryogels formed with the additives of URE or GHC testifies that the weakening influence of both of these chaotropes on the *E* values of the resultant PVACGs depends directly on the amount of URE or GHC in the samples, and this effect is similar at all the freezing temperatures over the range used in this study. As this took place at comparable molar concentrations of the additives, the weaker cryogels were those prepared in the presence of ionic GHC. This fact points to the higher ability of GHC to hinder PVA-PVA H-bonding in comparison to URE (this trend was also observed earlier [59]). At the same time, the character of the gel strength dependence on the cryogenic processing temperature for such URE- and GHC-containing PVACGs turned out to be unexpectedly different. Both types of curves had a bell-like shape, but in the former case, the curves were concave with minima points in the vicinity of *T*_cp_, around −30 °C (Figure 1b), while in the latter case, the curves were of a convex shape with the maxima points also near −30 °C (Figure 1d), i.e., these curves were like a “mirror reflection” of the curves for the URE-containing samples. 

In general, a convex-type bell-shaped dependence of sample rigidity on the temperature of its cryogenic preparation is known to be a characteristic of the PVACGs [5,30,31,43], and this fact is also illustrated by the curves for the additive-free samples in Figure 1b,d,f,h. A similar kind of dependence of the gel-formation “efficiency” on the *T*_cp_ is well-recognized to be the result of the competition between the promoting and hindering factors [3,5,30,43,46].

The key promoting ones are the cryoconcentrating effects. That is, when the initial liquid system does not freeze deeply, the solvent is incompletely crystallized, and the solutes are concentrated in the so-called unfrozen liquid microphase (UFLMP) [60], thus favouring gel-formation. In turn, the main hindering factors are a very high viscosity inside the UFLMP and a reduced temperature; these factors significantly slow down the translational and segmental mobility of the PVA chains [30,36,61,62,63].

In the case of the PVACGs formed in the presence of URE additives that caused a marked decrease of the mechanical toughness of the resultant gel samples with an increase in concentration of this chaotrope (Figure 1a), lowering the freezing temperature gave rise to the crystallization of a higher amount of solvent, and hence, this led to an increase in URE concentration in the UFLMP. Therefore, the Young’s moduli of the samples prepared at −25 and −30 °C were less than those of the URE-containing cryogels prepared at −20 °C (Figure 1b). In this context, some growth of the *E* values for the PVACGs formed by freezing at −35 °C needs some discussion. We suppose that such an effect is related to the solubility features of URE. Although the eutectic temperature of the binary system water-urea is known to lie near −9 °C [64], in the presence of a large amount of dissolved hydrophilic polymer (PVA in this case), which holds solvate non-freezable water, the eutectic point has to shift towards lower temperatures. Below −30 °C (e.g., at −35 °C) this point is obviously reached, and a portion of dissolved URE solidifies, thus decreasing the chaotrope concentration within the remaining unfrozen micro-regions of the macroscopically-frozen system. As a result, a lower URE concentration exerts interfering influence on the PVA cryotropic gel-formation. Of course, confirmation of such a mechanism requires additional future exploration. 

In addition, there are other questions such as the following: In the case of the PVA cryogels formed in the presence of the ionic GHC additives, and also chaotropic additives like the URE, why were the dependences *E* vs. *T*_cp_ of a convex shape, and why did an increase in the initial GHC concentration result in a “straightening” of the respective curves, i.e., their bend was reduced (Figure 1d)? Taking in account this tendency, it could be hypothesized that the respective “virtual” curves would be transformed to concave ones upon further increasing in the GHC concentrations, but, as we pointed out above, it was impossible, contrary to the URE-containing PVACGs (Figure 1a,b), to prepare the cryogel samples with a GHC content of 0.4 or 0.5 mol/L. 

In turn, for the PVA cryogels fabricated with kosmotropic additives, i.e., the non-ionic THL and the ionic HYP, the dependencies *E* vs. kosmotrope concentration (Figure 1e,g) and *E* vs. *T*_cp_ (Figure 1f,h) were respectively similar in their appearance. In the former case, the concave bell-like curves are characteristic of the influence of an increase in the amount of both organic and inorganic (inorganic salts) kosmotropic additives on the rigidity of the PVACGs prepared at various freezing temperatures [45,59]. The reason for this shape is the competition of two main oppositely directed trends. First, the increase in the THL or HYP concentration should, owing to their kosmotropic properties, facilitate H-bonding and thus favour PVA gelation and result in the formation of mechanically stronger cryogel samples. On the other hand, the higher the initial concentration of the low-molecular solute, the larger would be the volume of the UFLMP, owing to the additional solvate water. Hence, because of this “dilution”, the lower the PVA concentration will be therein, and weaker cryogels are formed at lower gelling polymer concentrations. As a result of the competition between these two trends, such bell-like dependences of the cryogels rigidity on the concentration of THL and HYP (Figure 1e,g), as well as other kosmotropes, [45,59] is observed.

In a view of the data on the character of both chaotropic (URE, GHC) and kosmotropic (THL, HYP) additives’ impact on the compression Young’s modulus of the respective PVACGs (Figure 1), it was also of interest to follow the similar influence of the same cryogels on the values of shear moduli measured under constant loading (Section 3.4). This was because of the known sensitivity of these mechanical characteristics of the gel materials in general [65,66,67] and PVA cryogels in particular [38,43,44,45,46] to the variations of the gel-formation conditions, so the required experiments have been carried out, and, as expected, the total character of the tendencies appeared to be virtually the same as for the *E* values with respect to both the additive-associated effects and freezing temperatures. However, certain differences have been revealed concerning the influence of these factors on the *G*_0_ and *G*_20_ moduli that are known to reflect, respectively, the elastic and plastic properties of the viscoelastic gel matrices [65]. In this context, the parameters that are capable of clearly showing the above differences turned out to be the relative values, i.e., *G*_0_^ac^/*G*_0_^af^ and *G*_20_^ac^/*G*_20_^af^, where ‘ac’ means additive-containing, and ‘af’ means additive-free. In addition, from our point of view, the table presentation (Table 1) rather than the plots-type presentation of the experimental data was more informative in this case. 

The analysis of these results revealed the following some rather interesting trends:(i)The ‘intensity’ of the increase or decrease of the ratios *G*_0_^ac^/*G*_0_^af^ and *G*_20_^ac^/*G*_20_^af^ depended on the chaotropic or kosmotropic properties of the additives, and such an ‘intensity’ was, as a rule, different with respect to the elastic and plastic characteristics of the PVACGs under study.(ii)This ‘intensity’ was also sensitive to the presence or absence of a charge in the molecules of a particular low-molecular additive.(iii)The character of the *G*_0_^ac^/*G*_0_^af^ and *G*_20_^ac^/*G*_20_^af^ variations with an increase in the additive amount was dependent on the cryogenic processing temperature (*T*_cp_) which was used for the PVACGs preparation.

For the URE- and GHC-containing cryogels, the values of these ratios markedly decreased with the growth of the chaotrope concentration, but certain differences were observed for PVACGs formed in the presence of non-ionic and ionic additives. Thus, if the *G*_0_ modulus of the URE-containing cryogels formed at −20 and −25 °C with an increase in URE concentration were lowered somewhat more in comparison with the *G*_20_ modulus of the same samples (**2a**–**f** and **3a**–**f**; Table 1), then for the PVACGs prepared by freezing at −30 and −35 °C the effect, say, was inverted (**4a**–**f** and **5a**–**f**; Table 1). In other words, in the former two cases the additives of this non-ionic chaotrope exerted more severe influence on the elastic properties of the respective PVA cryogels, whereas in the latter two cases the effect was on the plastic properties of these gel matrices. In turn, for the cryogels that contained the additives of the ionic chaotrope (i.e., GHC), the order of the effects was changed. Namely, for the PVACGs formed at −20 °C, the ‘intensity’ of lowering the *G*_20_^ac^/*G*_20_^af^ values with an increase in GHC concentration was weaker than for the *G*_0_^ac^/*G*_0_^af^ values (**6a**–**e**; Table 1); i.e., the plastic properties of these cryogels were affected to a lesser extent compared to the elastic properties. At the same time, for the samples prepared by freezing at lower temperatures (−25 or −30, or −35 °C), the opposite influence was observed (**7a**–**f, 8a**–**f** and **9a**–**f**; Table 1); i.e., the plastic properties of the respective PVACGs were affected to a higher extent than their elastic characteristics. 

The changes of the *G*_0_^ac^/*G*_0_^af^ and *G*_20_^ac^/*G*_20_^af^ ratios for the kosmotrope-containing PVA cryogels (the examples from **10a**–**e** to **17a**–**e** in Table 1) with increasing the additive concentration led to changes in the bell-like concave character of the curves that resembled the dependences for the *E* values (Figure 1e–h). With that, the influence of the kosmotropic non-ionic THL and ionic HYP on the elastic and plastic properties of the PVACGs prepared at different freezing temperatures was not identical. For instance, an increase in THL content in the samples formed at −20 °C (**10a**–**e**, Table 1) had a stronger effect on the elastic characteristics in comparison with the plastic ones: the *G*_0_^ac^/*G*_0_^af^ ratios were from 0.58 to 1.48, whereas the respective *G*_20_^ac^/*G*_20_^af^ values were higher and varied from 0.76 to 1.73. At the same time, for the PVACGs prepared at the lower freezing temperatures of either −25 °C (**11a**–**e**, Table 1) or −30 °C (**12a**–**e**, Table 1) or −35 °C (**13a**–**e**, Table 1), the trend was changed, and the plastic properties of the resultant cryogels were affected to a higher extent than the elastic ones. In the case of the HYP-containing PVACGs that were prepared via freezing at −20 or −25 °C (**14a**–**e** and **15a**–**e**, Table 1), an increase in the kosmotrope concentration registered a more severe influence with respect to the changes of the *G*_20_^ac^/*G*_20_^af^ ratios; i.e., HYP additives had a stronger effect on the plastic characteristics of the resultant gel matrices. However, when freezing at the lower temperatures of −30 or −35 °C was employed to produce the respective PVACGs (**16a**–**e** and **17a**–**e**
Table 1), the pattern was altered, and the elastic properties of the resultant cryogels were affected to a higher extent. 

Consequently, the data in Table 1 clearly testify that the additives of the low-molecular chaotropes and kosmotropes incorporated in the PVA feed solutions prior to their freeze-thaw processing exerted a rather sophisticated influence on the physico-mechanical properties of the resultant PVACGs. The final effects were stipulated by the ability of these additives to inhibit or promote the PVA-PVA intermolecular H-bonding by the particular chemical nature of the additive and its concentration, as well as by the freezing temperature used in the cryogenic process. 

### 2.2. Heat Endurance of PVACGs Prepared at Different Freezing Temperatures in the Absence or Presence of Chaotropic or Kosmotropic Additives

Yet other important physico-chemical characteristics of the non-covalent PVA cryogels are the temperature levels that these gel matrices can withstand without fusion upon heating the samples. So, the temperature of the gel-to-sol transition, i.e., the fusion temperature (***T*_f_**) of the PVACGs, can serve as an indicator of their heat endurance [5,30,31,68,69,70]. In this study, fusion temperatures were measured (Section 3.4) for the cryogels that were analogous to the samples whose physico-mechanical properties were collected in Figure 1 and Table 1. 

The results of the respective experiments are shown in Figure 2 as the dependences of the *T*_f_ values on the additive’s concentration (Figure 2a,c,e,g) and freezing temperature (Figure 2b,d,f,h). This is in accordance with the same logic scheme as the plots in Figure 1.

First of all, the graphs in Figure 2 clearly verify that an increase in the concentration of both chaotropes URE and GHC resulted in the progressive lowering of the *T*_f_ values (Figure 2a,c), while in the case of kosmotropic additives (i.e., THL and HYP), the effect was opposite—the heat endurance of these cryogels progressively increased (Figure 2e,g). The general runs of the curves that corresponded to the dependences of *T*_f_ versus additives concentration were very similar in the respective pairs of the chaotropes-containing and kosmotropes-containing cryogels, respectively. The intensity of the *T*_f_ value variations depended on the particular additive and on the freezing temperature employed for PVA cryotropic gel-formation. The comparison of the heat endurance inherent in the URE- and GHC-containing PVACGs prepared at equal initial concentrations of the respective chaotropes and at the same freezing temperatures shows that the latter cryogels, over the additive concentration range of 0.05–0.30 mol/L, had somewhat higher fusion temperatures (≈0.5–1.0 °C). The reason for such a result is assumed to be an increase in the ionic strength because of the GHC concentrating inside the UFLMP, thus inducing a salting-out-like effect with respect to PVA macromolecules. However, upon further growth of the GHC content (e.g., 0.4 or 0.5 mol/L), its chaotropic ability already exceeded the salting-out “power”, and the resultant GHC-containing cryogels were, as indicated in Section 2.1, too weak for reliable measurements of their physico-chemical characteristics. 

Concerning the dependences of the *T*_f_ values on the cryogenic processing temperature employed for the preparation of the chaotrope-containing PVACGs, some differences were found for the samples with the non-ionic URE and the ionic GHC additives. The former curves had a monotonously descending character with a lowering of the *T*_cp_ (Figure 2b), whereas the latter curves were mildly bell-shaped with the maxima at *T*_cp_ = −30 °C (Figure 2d). 

In the case of kosmotrope-containing PVACGs (Figure 2f,h), the respective also mildly bell-shaped temperature dependences did not virtually differ within the range of the non-ionic THL and ionic HYP concentrations from 0.1 to 0.5 mol/L. For the cryogels formed in the presence of a high concentration (1.0 mol/L) of both kosmotropes, the curves were similar and attested to the elevated heat endurance of such gel samples. Evidently, this effect owes to the strong promotion of the H-bonding processes induced by the elevated concentrations of these kosmotropes, because the higher the amount of H-bonds in the physical junction knots of a 3D polymeric network, the higher the quantity of energy required for the dissociation of such intermolecular links [69,71,72].

As a whole, the data of Figure 2 show that thermal properties of PVACGs prepared and explored in this research are mainly affected by the type and the concentration of the chaotropic or kosmotropic additives rather than by the freezing temperature used in the course of these cryogels’ formation. At least this is so over the *T*_cp_ value range from −20 to −35 °C. 

Herewith, the character of the cryogels’ fusion temperature changes with additive concentration for the chaotrope-containing samples (Figure 2a,c) was qualitatively close to the respective dependences of the mechanical moduli (Figure 1a,c; examples 2–5 and 6–10 in Table 1). In these cases, with an increase in the initial URE or GHC amount in the PVA solution to be structured cryogenically, both the rigidity and heat endurance of the resultant cryogels progressively decreased. At the same time, for the kosmotrope-containing PVACGs, the character of the variation in the analogous dependences was distinct. If with an increase in the THL or HYP concentration the values of the mechanical moduli of the resultant cryogels passed through the minima (Figure 1e,g; examples 11–13 and 14–17 in Table 1), the *T*_f_ values grew monotonically (Figure 2e,g). 

In this context, it should be pointed out that characteristics such as a cryogel’s fusion temperature is almost insensitive to the microstructure of the gel samples, i.e., to their total porosity, and the amount, size and shape of the macropores. In contrast, the physico-mechanical properties of these heterophase PVA cryogels are sensitive to the texture of polymeric matrices [43,44]. In fact, this is because the macropores in PVACGs are the dispersed defects within the bulk material and thus are capable of substantially influencing the integral toughness. Therefore, this information, especially the quantitative results about the macroporous morphology of different PVA cryogels prepared under various conditions, is of great significance.

### 2.3. Morphometric Analysis of the Structural Features of PVACGs Prepared in Absence and Presence of Chaotropic or Kosmotropic Additives

The micrographs in Figure 3 show typical images of the macroporous morphology of the PVA cryogels under the study. All of these samples, namely the additive-free as well as the chaotrope- and the kosmotrope-containing ones, have been prepared via freezing the initial solutions at −30 °C. With that, the concentration of URE, GHC, THL, and HYP in the additive-containing samples was equal to 0.2 mol/L. 

The characteristic feature of these images and those obtained for the cryogel samples, the physicochemical properties of which are summarized in Figure 1 and Figure 2 and Table 1, is the alternation of the light (water-filled macropores) and dark (polymeric gel walls of macropores) structural elements. The average cross-section of the macropores in the cryogels under study was over the range of 2–10 μm. Such a pattern is typical for PVA cryogels in general [43,44,45,46,47,59]. In this study, subsequent morphometric analysis was performed using images of similar to those in Figure 3.

Various neural network architectures were used to classify multiple images of cryogels VGG-16, InceptionV3, ResNet50, T-CNN-3, and T-CNN-2. The dataset of images was divided into a different number of classes: 2, 3, 4, 5, 8, 12, 16, 20, 36. In part of the experiments, the ImageNet dataset was used to pre-train neural networks. It was found that the most accurate results were obtained for a small number of classes, from 2 to 5, while for a number of classes greater than 5, the accuracy values did not exceed 0.5–0.65. As the number of classes increased, the average accuracy decreased, and for some classes it could be about 0.1. 

Based on this, the dataset from a set of images should be formed based on the additive type (3 classes of the images answering to the PVA cryogels prepared without foreign additives and in presence of either kosmotropes, or chaotropes).

Despite mediocre results for reference datasets with object images, the T-CNN architecture performed well in classifying texture images. Based on a comparative analysis of the results of various T-CNN modifications [73] we can conclude that the T-CNN-2 and T-CNN-4 architectures performed better.

To test the implementation of a convolutional network with the T-CNN-2 architecture, 12 classes of the Kylberg reference dataset were taken [74]. The classification results showed an excellent solution to the problem with an accuracy classification of about 100% for 12 classes. To evaluate the results obtained, cross-validation [75] with splitting the original Kylberg dataset into several balanced parts was used. The parts were formed randomly. The dynamics of accuracy during neural network training for five independent launches are shown in Figure 4. According to the results of the test, the average accuracy did not decrease and remained at the level of about 100%. The results of cross-validation confirm that the implementation of the T-CNN-2 architecture works when classifying the texture images.

From the accuracy dynamics in Figure 4, it can be seen that high accuracy is already achieved by the 30th epoch of training. This is typical for all training sets formed during cross-validation. Using the Kylberg dataset allows us to fine-tune the parameters of a neural network with the T-CNN-2 architecture, and it also allows us to determine with high accuracy whether a precedent set belongs to a class. A dataset was formed of only three classes of cryogel sections with various types of additives: kosmotropes, chaotropes, and without additives. For training, 48 thousand precedents (fragments) were taken, 4800 precedents each in the validation and test set. The training was conducted over 100 epochs with the possibility of an early stop based on the results of the validation set. When considering the classification results by class (Table 2), it can be seen that the results for the image class ‘Kosmo’ are better than for the other classes. However, it is necessary to take into account that all results are satisfactory, since the values of the Recall and F1-mesure metrics are quite low.

To avoid the specificity of the cryogel section images, a new dataset was formed as a balanced combination of 12 classes of the Kylberg dataset and three classes of the images. It was assumed that training a neural network on a dataset with images from Kylberg will allow us to configure the weights of the neural network better and faster. A number of computational experiments were performed with various parameters of neural network training.

The data in Table 3 show significant improvements in the values of the quality metrics for classifying test cases relative to previous classification attempts without a combination with 12 classes of the Kylberg dataset (for example, the value of the F1-measure for the ‘Kosmo’ class increased significantly to 0.80 against 0.05 in previous attempts, and the Recall of the ‘Clear’ class increased to 1). From the results shown, it can be seen that the cases of the ‘Kosmo’ and the ‘Clear’ classes are defined fairly well. In turn, the definition of the ‘Chao’ class precedents is not reliable and deserves a “satisfactory” quality assessment. With this number of 15 classes, high accuracy values should not be considered separately as evidence of an excellent classification. The dynamics of accuracy during training Figure 5 also confirms the quality of classification results.

Large accuracy values for the validation set, comparable to the accuracy values for the training set, confirm the success of training. When divided into classes based on the presence of additives with the participation of 12 classes of the Kylberg dataset, good results were also obtained.

The results of this experiment can be considered to be good, because with an accuracy of 0.98, the F1-measure values for both classes are about 0.9. The results of computational experiments have shown that by using a neural network of the T-CNN-2 architecture with a combined dataset (Kylberg texture dataset and PVACG dataset), it was possible to classify images of PVACG by the type of additives (class 3) and by the presence of additives (class 2).

## 3. Materials and Methods

### 3.1. Materials 

The following compounds were used in the experiments without additional purification: PVA with molecular weight of ~86000 Da and 100% degree of deacetylation (Acros Organics, Morris Plains, NJ, USA), URE (ultra-pure) (Sigma, Ronkonkoma, NY, USA), GHC (>99.5%) (Helicon, Moscow, Russian Federation), THL (>98%) (Panreac, Barcelona, Spain), HYP (>98.5%) (Rexim, Paris, France), Congo red dye (Aldrich, St. Louis, MO, USA) as well as gelatine (photo quality), phenol (pure for analysis) and glycerol (pure for analysis) (all from Reakhim Co., Moscow, Russian Federation). 

### 3.2. Preparation of PVA Solutions with and without the Soluble Low-Molecular Additives 

PVA solutions were prepared as reported elsewhere [38,43,44,45,46,47]. In brief, the necessary amount of dry PVA powder was suspended in the required volume of water to give a PVA concentration of 100 g/L in the solution to be prepared. The system was stored for 18 h at room temperature for PVA swelling. Then the suspension was heated for 50 min under stirring on a boiling water bath to dissolve the polymer completely. The sample was weighed before and after heating, and the amount of evaporated water was added. Upon the preparation of PVA solutions that additionally contained a chaotropic or kosmotropic substance, the required amount of the particular substance was dissolved in the polymer-containing solution. Prior to the freeze-thaw processing the feed solutions were treated for 20 min at room temperature in the ultrasonic bath UNITRA (Unitra, Olsztyn, Poland) for the removal of air bubbles.

### 3.3. Preparation of Cryogel Samples

The PVACG samples preparation was carried out essentially in accordance with the previously described procedure. The cryogels for physicomechanical measurements were moulded in sectional duralumin containers (inner dia. 15 mm, height 10 mm), and the samples for the measurement of their fusion temperature were prepared in transparent polyethylene test tubes (inner dia. 10 mm) [43,44,45,46,47,49]. In the last case, the tubes were filled with 5 mL of polymer solution, and a stainless steel ball with the dia. of 3.5 mm and weight of 0.275 ± 0.005 g was placed on the bottom of each test tube. The containers and the tubes were put into the chamber of a precision programmable cryostat FP 45 HP (Julabo, Seelbach, Germany) and incubated at a pre-set sub-zero temperature for 12 h. Then, the temperature was raised to 20 °C at the rate of 0.03 °C/min as governed by the cryostat microprocessor.

### 3.4. Physicochemical Characteristics of PVACG Samples

The following physico-mechanical characteristics for the prepared gel samples were evaluated as described elsewhere for different additive-free and additive-containing PVACGs. The apparent instantaneous shear modulus (*G*_0_) and shear modulus for the 20-min-loading (*G*_20_) were measured by the penetration method at a constant load of 4.9·10^−3^ N using a Kargin-Sogolova dynamometric balance [38,43,44,45,46]. The values of compression Young’s modulus (*E*) were determined from the linear portion of the stress-strain dependence found by using a TA-Plus automatic texture analyser (Lloyd Instruments, West Sussex, UK) at a loading rate of 0.3 mm/min until 30% deformation [47,59]. 

The gel fusion temperatures (***T*_f_**) of the PVACG samples were measured using the already described procedure [38,43,44,45,46,47,59]. In brief, the tightly corked polyethylene tube in which the cryogel had been prepared with a metal ball at the bottom was placed upside down into the water bath and heated at the rate of 0.4 °C/min. The gel fusion point was determined as the temperature when the ball fell down onto the stopper of the test-tube after passing through the fused matter. The moduli and fusion temperatures were measured for three parallel samples; the samples were prepared in 3–5 independent experiments. The results obtained were averaged.

### 3.5. Optical Microscopy of PVACGs

Studies of the PVACGs microstructure were performed with optical microscopy. Thin gel sections were contrasted by Congo red staining and prepared according to the earlier reported technique [47,59]. The sections of ~10 µm thickness were cut in a direction orthogonal to the axis of the cylindrical samples using cryomicrotome SM-1900 (Leica, Wetzlar, Germany); the microscopy investigations were carried out with an Eclipse 55i (Nikon, Tokyo, Japan) instrument equipped with a digital camera. 

### 3.6. Morphometric Analysis of the Microscopy Images

#### 3.6.1. Classification

Many images of cryogels were obtained under different conditions, and they can be initially classified according to several characteristics:(i)The presence of additives (two classes: there are additives, no additives).(ii)The type of additives (three classes: kosmotropes, chaotropes, no additives).(iii)Particular additives (five classes: four substances and without additives).(iv)Concentration of the additives (two classes).(v)Cryogenic processing temperature (four classes: −20 to −35 with 5 degree steps)

With various divisions into classes based on these characteristics, from two to 40 classes may be obtained.

#### 3.6.2. Creating a Dataset

When solving the problem of classifying textures in images for a neural network, image fragments are represented, since the source images can have a large size of 1920 × 1080 (Full HD), 4096 × 3072 (4 K) or even more. At the same time, the input layer of neural networks will have a dimension on each side of about 200–500 points.

Let’s describe the method of generating a dataset for training a neural network: A total of 654 images of cryogel sections were obtained using optical microscopy methods. The resulting images were of the same size: 2272 × 1704 points. Since the images were not full-color (red dye), it was decided to use only the colour intensity (1 channel). In order to preserve the full information of the images, zooming was not used. To use all of the information in the images, each source image was divided into fragments of 256 × 256 points with 50% overlap at each coordinate. Since the orientation of the texture on the image is unknown, each image must be represented from a set of fragments rotated at different angles. During the process of augmenting multiple images, the original images were rotated at angles from 15 to 345 degrees in 15-degree increments. The fragments of the same size and with the same overlap were highlighted on the rotated images. As a result of fragment selection and augmentation of the original set of 654 images, a dataset consisting of 4,559,688 fragments was formed. The size of the dataset on disk was about 61 GB. When performing the training, validation, and test sets, it was taken into account that fragments of a single image can only be located in one of the sets.

#### 3.6.3. Neural Network Architecture

The T-CNN architecture, which has different configurations from T-CNN-1 to T-CNN-5, was used. Its configurations differ only in the number of convolution layers. In the article [75], the results of testing all of these configurations on reference datasets are presented. The T-CNN-2 parameters used are summarized in Table 4.

After each layer, except for the fully connected ones, batch normalization was used, and the coefficient of random disconnection of neurons was 0.4. The initial weights of neural networks were taken after the pre-training stage on the ImageNet reference set. During training, the Adam learning algorithm was used with the criterion of stopping early training when using a validation subset.

Neural networks were implemented in the Python 3.6 programming language using TensorFlow and Keras. The computational experiments on training neural networks were conducted on a personal computer equipped with a Core-i7 8550U CPU/8 GB RAM/NVidia MX150 2 GB (CUDA 10.2).

#### 3.6.4. Classification Quality Metrics

The following metrics are used to evaluate the quality of classification:Accuracy;Precision;Recall;F1-measure.

These metrics can be used for any number of classes. For an explanation, let’s take the binary classification. The binary classifier results in four sets of results. This includes errors in defining two classes, False Negative (FN) and False Positive (FP), and the correct definition of true precedent classes, True Negative (TN) and True Positive (TP). The metrics used are combinations of the sizes of these 4 sets. Accuracy is the ratio of correctly defined precedents of both classes to the total number of precedents. Precision is the ratio of correctly defined Positive class precedents to the total number of class precedents defined by the classifier: TP/(TP+FP). Recall defines the proportion of correctly defined Positive class precedents to the total number of class precedents: TP/(TP+FN). F1-measure is the average harmonic of Precision and Recall. All of the listed metrics take values on the segment [0,1]. The metric value is better the closer it is to 1. When evaluating the classification quality, metrics should be considered together, not separately. In general, if the number of classes is more than 2, the training or test set will most likely not be balanced and the average Accuracy value will not determine the classification quality.

## 4. Conclusions

In this study, macroporous poly(vinyl alcohol) cryogels were prepared at different temperatures when the initial polymer solutions were either additive-free or contained chaotropic (urea, guanidine hydrochloride) or kosmotropic (trehalose, hydroxyproline) additives. After completion of gelation, the rigidity and heat endurance of the resultant cryogels were evaluated, and their macroporous structure was investigated using optical microscopy. The images obtained were treated mathematically, and deep neural networks were used for the classification of such images. Training as well as test sets were used for the classification.

The use of the GCN-2 architecture was successful in solving the problem of classifying images of cryogels based on signs of the presence of an additive and its type. Forming a dataset for training a neural network in conjunction with the Kylberg texture dataset and using a small number of classes improved the classification results. It should be noted the best image recognition was for PVACG without additives and with kosmotrope additives. The results of the study can be used to analyse and classify other sets of images of porous materials. The presented classification results illustrate that the type of the additive introduced in the initial PVA solution is the most significant factor when separating the presented texture dataset, but for other texture datasets, the set of essential morphometric features for separating the dataset into classes may be different. It means that for the prediction of the conditions for the preparation of such cryogels with desirable macroporous morphology, it is necessary to classify images of PVA cryogel sections by a larger number of morphometric features. Solving the classification problem for a larger number of classes and a larger number of the PVA cryogels morphological features is the main direction of further research, which is in progress now.

## Figures and Tables

**Figure 1 molecules-25-04480-f001:**
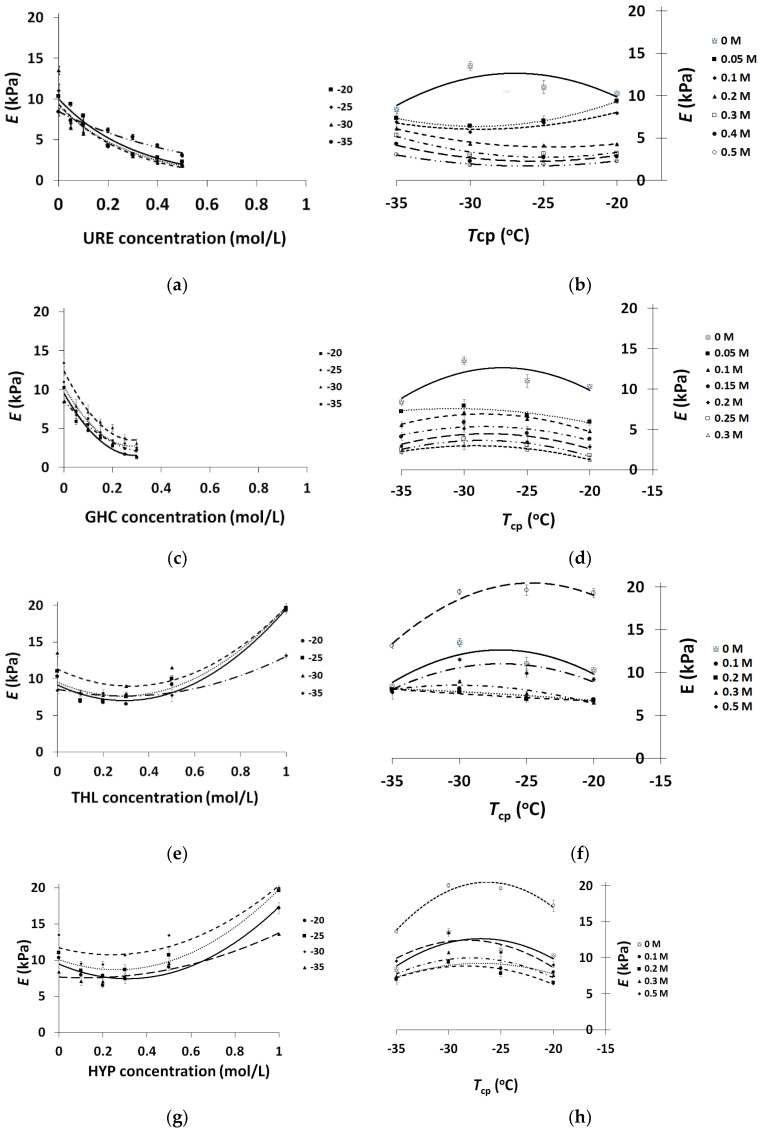
Dependences of the Young’s modulus (*E*) of PVACGs on the concentration of chaotropic or kosmotropic additives in the feed solutions (**a**,**c**,**e**,**g**) and the cryogenic processing temperature (**b**,**d**,**f**,**h**).

**Figure 2 molecules-25-04480-f002:**
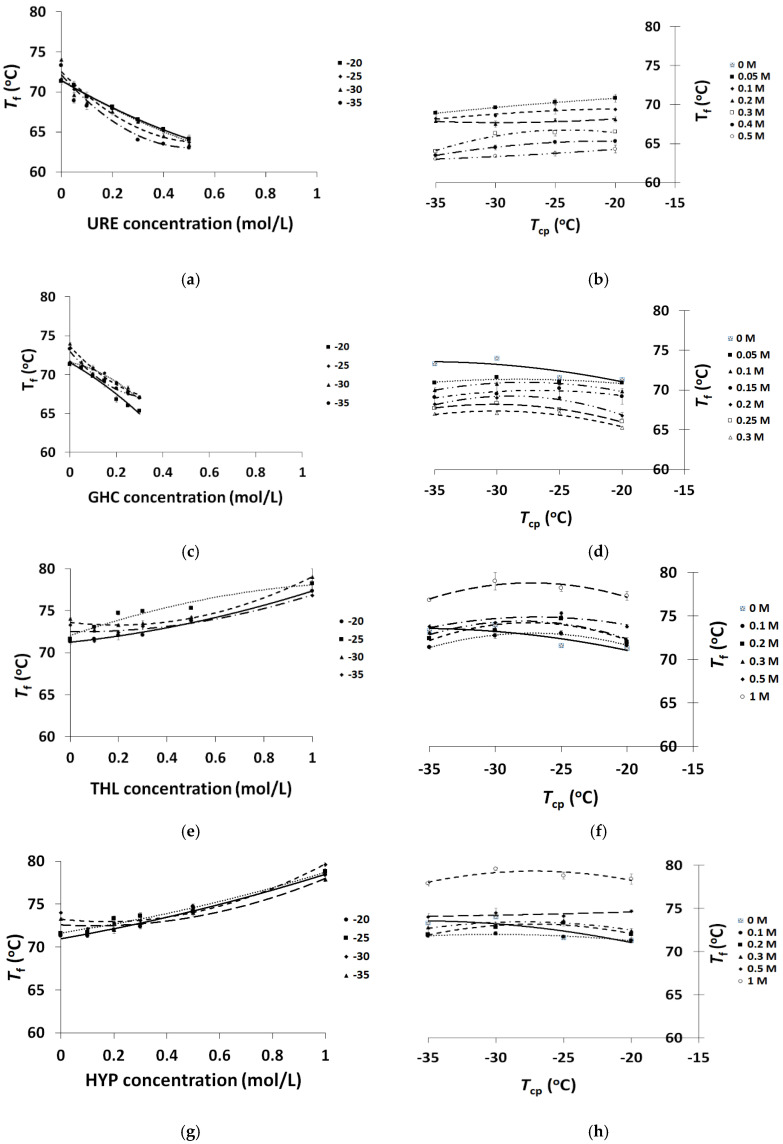
Dependences of the fusion temperature (*T*_f_) of PVACGs on the concentration of chaotropic or kosmotropic additives in the feed solutions (**a**,**c**,**e**,**g**) and the cryogenic processing temperature (**b**,**d**,**f**,**h**).

**Figure 3 molecules-25-04480-f003:**
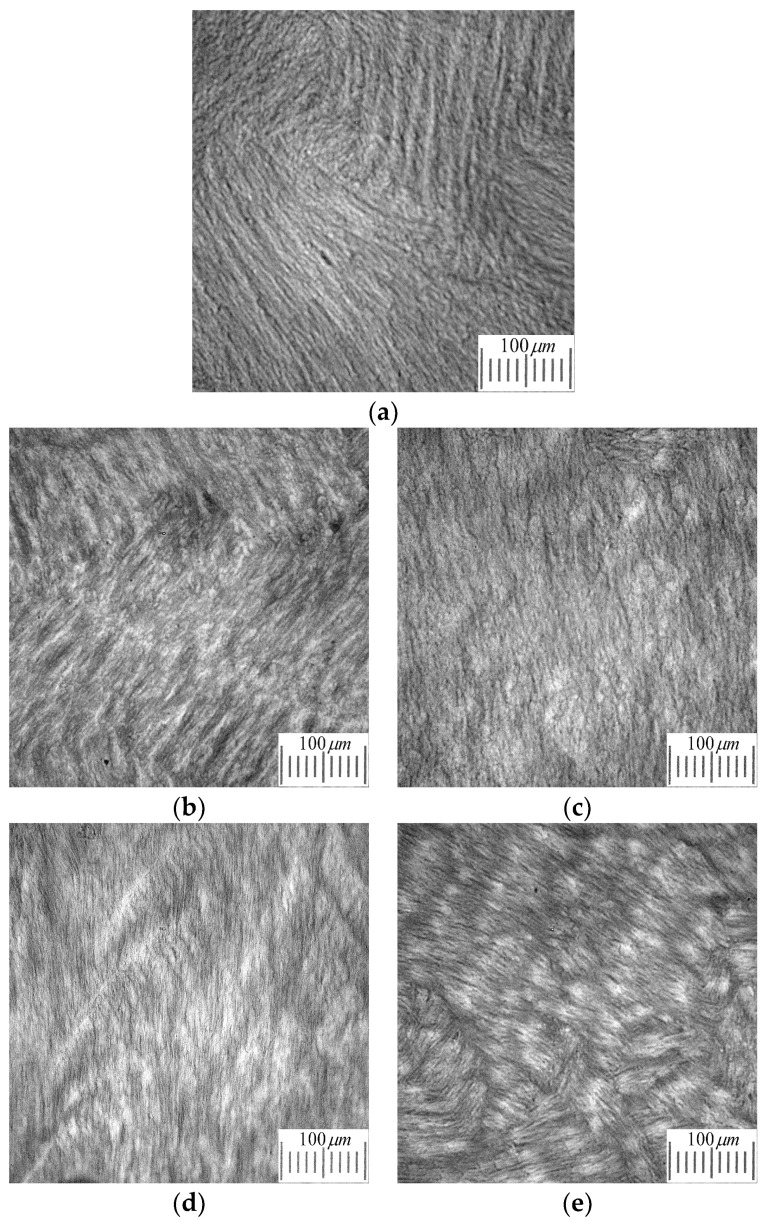
Black-and white micrographs of the thin sections of PVA cryogels prepared from the PVA aqueous solution (100 g/L) without additives (**a**) and containing 0.2M URE (**b**), 0.2M GHC (**c**), 0.2M THL (**d**) and 0.2M HYP (**e**) (cryogenic processing temperature in all the cases was −30 °C).

**Figure 4 molecules-25-04480-f004:**
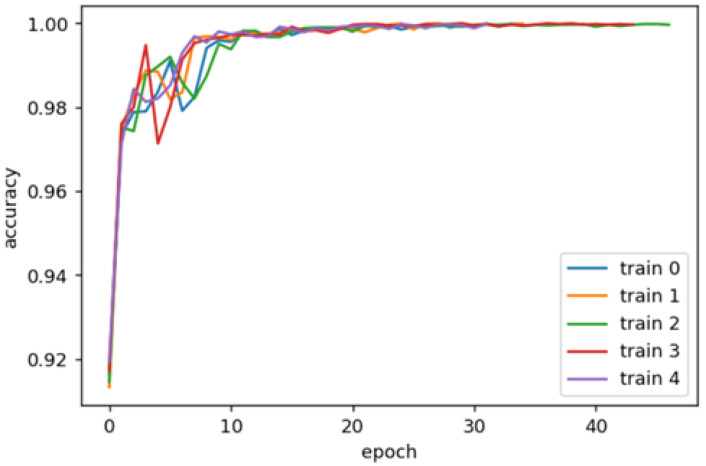
Accuracy dynamics for five independent TCN-2 learning attempts in 12 classes of the Kylberg dataset.

**Figure 5 molecules-25-04480-f005:**
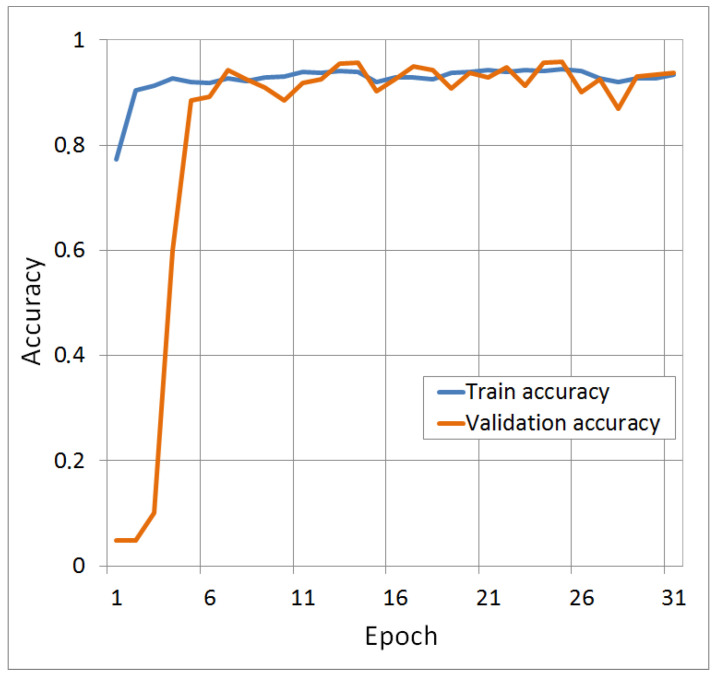
Accuracy dynamics for training and validation sets during training.

**Table 1 molecules-25-04480-t001:** Shear moduli of the PVACG samples.

	Composition of Feed Solution						Physico-Mechanical			
Example Number		Low-Molecular Additive					Parameters			
	PVA	Type	Charge	Substance	Concentration (mol/L)	*T* _cp_ ^a^	*G* _0_	*G*_0_^ac^/*G*_0_^af^	*G* _20_	*G*_20_^ac^/*G*_20_^af^
	(g/L)					(°C)	(kPa)		(kPa)	
**1a**	100	-	-	none	-	−20	9.13 ± 0.40	-	5.77 ± 0.33	-
**b**		-	-		-	−25	10.0 ± 0.80	-	6.20 ± 0.30	-
**c**		-	-		-	−30	12.8 ± 0.30	-	11.1 ± 0.30	-
**d**		-	-		-	−35	9.03 ± 0.21	-	6.94 ± 0.10	-
**2a**		chao- trope	non- ionic	URE	0.05	−20	6.41 ± 0.25	0.70	5.05 ± 0.11	0.88
**b**					0.10		5.86 ± 0.16	0.64	4.93 ± 0.13	0.85
**c**					0.20		3.17 ± 0.17	0.35	2.90 ± 0.08	0.50
**d**					0.30		1.93 ± 0.13	0.21	1.29 ± 0.21	0.22
**e**					0.40		1.67 ± 0.11	0.18	ND	-
**f**					0.50		1.42 ± 0.12	0.16	ND	-
**3a**					0.05	−25	6.21 ± 0.69	0.62	4.33 ± 0.41	0.70
**b**					0.10		4.76 ± 0.10	0.48	3.79 ± 0.13	0.61
**c**					0.20		3.54 ± 0.26	0.35	2.60 ± 0.28	0.42
**d**					0.30		2.21 ± 0.19	0.22	1.58 ± 0.30	0.25
**e**					0.40		1.65 ± 0.05	0.17	1.35 ± 0.13	0..22
**f**					0.50		1.15 ± 0.09	0.12	1.04 ± 0.14	0.17
**4a**					0.05	−30	5.01 ± 0.23	0.39	3.67 ± 0.39	0.33
**b**					0.10		4.31 ± 0.25	0.34	3.60 ± 0.48	0.32
**c**					0.20		3.14 ± 0.08	0.25	2.55 ± 0.15	0.23
**d**					0.30		2.01 ± 0.15	0.16	1.50 ± 0.12	0.14
**e**					0.40		1.57 ± 0.05	0.12	1.11 ± 0.15	0.10
**f**					0.50		0.90 ± 0.10	0.07	ND	-
**5a**					0.05	−35	8.67 ± 0.15	0.96	6.04 ± 0.20	0.87
**b**					0.10		7.56 ± 0.80	0.83	4.93 ± 0.11	0.71
**c**					0.20		6.25 ± 0.11	0.69	3.93 ± 0.19	0.57
**d**					0.30		3.73 ± 0.09	0.41	2.80 ± 0.20	0.40
**e**					0.40		3.05 ± 0.11	0.34	1.79 ± 0.13	0.26
**f**					0.50		2.55 ± 0.15	0.28	1.71 ± 0.21	0.24
**6a**			ionic	GHC	0.05	−20	5.57 ± 0.09	0.61	3.64 ± 0.08	0.63
**b**					0.10		3.83 ± 0.21	0.42	2.54 ± 0.08	0.44
**c**					0.15		2.55 ± 0.07	0.28	1,79 ± 0.09	0.31
**d**					0.20		2.06 ± 0.06	0.23	1.44 ± 0.12	0.25
**e**					0.30		ND	-	ND	-
**7a**					0.05	−25	7.35 ± 0.05	0.74	4.49 ± 0.47	0.72
**b**					0.10		6.43 ± 0.09	0.64	3.35 ± 0.39	0.54
**c**					0.15		3.86 ± 0.14	0.39	2.05 ± 0.31	0.33
**d**					0.20		2.79 ± 0.07	0.28	1.49 ± 0.19	0.24
**e**					0.25		2.00 ± 0.12	0.20	1.20 ± 0.12	0.19
**f**					0.30		1.48 ± 0.04	0.15	ND	-
**8a**	100	chao- trope	ionic	GHC	0.05	−30	8.08 ± 0.46	0.63	5.50 ± 0.18	0.50
**b**					0.10		7.11 ± 0.49	0.56	3.48 ± 0.40	0.31
**c**					0.15		5.55 ± 0.51	0.43	2.88 ± 0.16	0.26
**d**					0.20		4.20 ± 0.38	0.33	2.65 ± 0.15	0.24
**e**					0.25		2.76 ± 0.20	0.22	1.86 ± 0.16	0.17
**f**					0.30		2.29 ± 0.19	0.18	1.52 ± 0.18	0.14
**9a**					0.05	−35	6.61 ± 0.09	0.73	4.65 ± 0.21	0.67
**b**					0.10		4.52 ± 0.06	0.50	3.12 ± 0.30	0.45
**c**					0.15		3.46 ± 0.04	0.38	2.59 ± 0.21	0.37
**d**					0.20		2,29 ± 0.13	0.25	1.61 ± 0.25	0.23
**e**					0.25		1.59 ± 0.11	0.18	1.25 ± 0.15	0.18
**f**					0.30		1.22 ± 0.12	0.14	ND	-
**10a**		kosmotrope	non-ionic	THL	0.1	−20	5.25 ± 0.59	0.58	4.40 ± 0.12	0.76
**b**					0.2		4.85 ± 0.05	0.53	4.21 ± 0.07	0.73
**c**					0.3		5.65 ± 0.15	0.62	4.38 ± 0.14	0.76
**d**					0.5		7.40 ± 0.10	0.81	5.30 ± 0.10	0.92
**e**					1.0		13.5 ± 0.60	1.48	10.0 ± 0.20	1.73
**11a**					0.1	−25	7.40 ± 0.60	0.74	3.90 ± 0.10	0.63
**b**					0.2		6.11 ± 0.29	0.61	3.70 ± 0.10	0.60
**c**					0.3		7.32 ± 0.08	0.73	4.40 ± 0.12	0.71
**d**					0.5		10.4 ± 0.20	1.04	6.04 ± 0.08	1.03
**e**					1.0		16.7 ± 0.50	1.67	7.60 ± 0.10	1.23
**12a**					0.1	−30	5.69 ± 0.19	0.44	5.27 ± 0.09	0.47
**b**					0.2		5.80 ± 0.12	0.45	4.65 ± 0.21	0.42
**c**					0.3		7.03 ± 0.21	0.55	4.51 ± 0.15	0.41
**d**					0.5		11.7 ± 0.20	0.91	5.30 ± 0.28	0.45
**e**					1.0		16.9 ± 0.90	1.32	10.0 ± 0.20	0.90
**13a**					0.1	−35	7.88 ± 0.12	0.87	4.39 ± 0.21	0.63
**b**					0.2		6.95 ± 0.45	0.77	3.07 ± 0.09	0.44
**c**					0.3		6.90 ± 0.52	0.76	2.56 ± 0.40	0.37
**d**					0.5		8.01 ± 0.87	0.89	4.39 ± 0.11	0.63
**e**					1.0		15.0 ± 0.20	1.66	8.03 ± 0.29	1.16
**14a**			ionic	HYP	0.1	−20	6.17 ± 0.32	0.68	4.80 ± 0.50	0.83
**b**					0.2		5.03 ± 0.35	0.55	4.65 ± 0.13	0.81
**c**					0.3		5.98 ± 0.12	0.65	5.10 ± 0.22	0.88
**d**					0.5		9.35 ± 1.03	1.02	7.00 ± 0.25	1.21
**e**					1.0		12.4 ± 0.10	1.36	10.5 ± 0.50	1.82
**15a**					0.1	−25	8.00 ± 0.12	0.80	5.05 ± 0.25	0.81
**b**					0.2		6.68 ± 0.16	0.67	4.65 ± 0.11	0.75
**c**					0.3		7.62 ± 0.10	0.76	5.27 ± 0.13	0.85
**d**					0.5		10.7 ± 0.30	1.07	7.94 ± 0.32	1.28
**e**					1.0		16.2 ± 0.40	1.62	12.4 ± 0.20	2.00
**16a**	100	kosmotrope	ionic	HYP	0.1	−30	8.05 ± 0.21	0.63	5.27 ± 0.13	0.47
**b**					0.2		7.60 ± 0.40	0.59	4.93 ± 0.21	0.44
**c**					0.3		9.30 ± 0.22	0.73	6.04 ± 0.20	0.54
**d**					0.5		13.5 ± 0.40	1.05	9.08 ± 0.18	0.82
**e**					1.0		18.1 ± 0.30	1.41	14.0 ± 0.20	1.26
**17a**					0.1	−35	6.71 ± 0.11	0.74	4.50 ± 0.40	0.65
**b**					0.2		6.62 ± 0.08	0.73	4.48 ± 0.12	0.65
**c**					0.3		7.90 ± 0.60	0.87	4.65 ± 0.10	0.67
**d**					0.5		8.40 ± 0.88	0.93	6.29 ± 0.11	0.91
**e**					1.0		11.5 ± 0.50	1.27	6.80 ± 0.20	0.98

^**a**^ The data for the case of *T*_cp_ = −20 °C are from ref. [44]. ND—non-determined (too weak sample for a reliable measurement).

**Table 2 molecules-25-04480-t002:** Classification results for 3 classes of PVACG without Kylberg classes in learning set.

Name	Kosmo	Chao	Clear
Accuracy	0.80	0.81	0.79
Precision	0.65	0.79	0.70
Recall	0.87	0.59	0.63
F1-measure	0.74	0.68	0.67

**Table 3 molecules-25-04480-t003:** Classification results for 3 classes of PVACG with 12 Kylberg classes in learning set.

Name	Kosmo	Chao	Clear
Accuracy	0.97	0.95	0.98
Precision	0.84	0.65	0.79
Recall	0.76	0.58	1.00
F1-measure	0.8	0.61	0.88

**Table 4 molecules-25-04480-t004:** Classification results for two classes of PVACG ***with*** 12 Kylberg classes in learning set.

Name	Kosmo + Chao	Clear
Accuracy	0.98	0.98
Precision	0.93	0.84
Recall	0.86	0.96
F1-measure	0.89	0.89

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
