# Peer review of "Cryostructuring of Polymeric Systems : Application of Deep Neural Networks for the Classification of Structural Features Peculiar to Macroporous Poly(vinyl alcohol) Cryogels Prepared without and with the Additives of Chaotropes or Kosmotropes"

_molecules, 2020, doi:10.3390/molecules25194480_

Round 1

Reviewer 1 Report

In this work, macroporous poly(vinyl alcohol) cryogels were prepared at different temperatures when the initial polymer solutions were either additive-free or contained chaotropic (urea, guanidine, hydrochloride) or kosmotropic (trehalose, hydroxyproline) additives. The work is interesting and will attract the interests from the field. The authors are suggested to enlarge the labels in the figure to make them better presented.

Author Response

Thanks a lot for your positive, in whole, evaluation of this article.

In accordance with the reviewer’s remarks, all labels in the figures have been enlarged.

Reviewer 2 Report

Reviewers comments to the author

The manuscript presents the effects of additives on PVA cryogel properties. The paper is very interesting, the presented studies are well planned, systematically investigated, and analyzed. The Authors did a nice job of explaining the concepts, approaches, and rationale at every step. The manuscript is clearly written.

General comments

  1. The authors mention that ‘the results of the study can be used to analyze and classify other sets of images of porous materials.’ Please elaborate on the scope. How the observations can be related to the behavior and structural features of other physically cross-linked gels in presence of chaotropic or kosmotropic additives.

  1. Multiple factors play a role in deciding the morphological features of cryogels such as polymer composition, crosslinking density, fabrication temperature, additives etc. The morphological features and mechanical properties are important while deciding the suitability of fabricated cryogel for a particular application. Do you think for a studied cryogel-additive system, is it possible to predict the fabrication conditions for getting a cryogel with desired features?

  1. The quality of some figures should be improved. Please ensure that the figure legends are readable.

Author Response

General comments

  1. The authors mention that ‘the results of the study can be used to analyze and classify other sets of images of porous materials.’ Please elaborate on the scope. How the observations can be related to the behavior and structural features of other physically cross-linked gels in presence of chaotropic or kosmotropic additives.

Thank you for your positive review of this article.

The method of processing PVA cryogel section images and the method of classifying PVA cryogel section images using convolutional neural networks presented in this paper are quite universal and can be used for other texture images of porous materials. The presented classification results illustrate that the type of the additive introduced in the initial PVA solution is the most significant when separating the presented texture dataset (we added these comments in the ‘Conclusions’ of the revised version of the manuscript).

  1. Multiple factors play a role in deciding the morphological features of cryogels such as polymer composition, crosslinking density, fabrication temperature, additives etc. The morphological features and mechanical properties are important while deciding the suitability of fabricated cryogel for a particular application. Do you think for a studied cryogel-additive system, is it possible to predict the fabrication conditions for getting a cryogel with desired features?

In this paper, the principal possibility of successful classification only by the type of low-molecular additive was shown. But for other texture datasets, the set of essential morphometric features for separating the dataset into classes may be different. It means that for the prediction of the conditions for the preparation of such cryogels with desirable macroporous morphology, it is necessary to classify images of PVA cryogel sections by a larger number of morphometric features. Solving the classification problem for a larger number of classes and a larger number of the PVA cryogels morphological features is the main direction of further research, which is in progress now (we also added these comments in the ‘Conclusions’ of the revised version of the manuscript).

  1. The quality of some figures should be improved. Please ensure that the figure legends are readable.

In accordance with the reviewer’s remarks, all labels in the figures have been enlarged.